

# Perception and practices during the COVID-19 pandemic in an urban community in Nigeria: a cross-sectional study

Olayinka Ilesanmi[1,2] and Aanuoluwapo Afolabi[1]

[1] Department of Community Medicine, University of Ibadan, Ibadan, Oyo State, Nigeria
[2] Department of Community Medicine, University College Hospital, Ibadan, Oyo State, Nigeria

## ABSTRACT

**Background:** Various perceptions and practices have been associated with the COVID-19 pandemic. In this study, we assessed the perception and practices regarding COVID-19 among residents in selected urban communities of Ibadan, Oyo State, Nigeria.

**Methods:** A descriptive cross-sectional study design using a multi-stage sampling technique was used to recruit 360 respondents (Mean age: 33.2 ± 10.6 years; 62.5% females) from households in Ibadan. Data were collected using an interviewer-administered questionnaire from 3rd to 6th June 2020. Those who demonstrated washing of the palm, back of the hand, spaces between the fingers, fingernails, wrist, and thumbs had six points and were categorized to have had a good practice of handwashing. Descriptive statistics were conducted. Bivariate analyses of sociodemographic characteristics and good handwashing practices were conducted using Chi-square test. Logistic regression was conducted to identify the determinants of good handwashing practices. $P$-values < 0.05 were statistically significant.

**Results:** Going to the hospital (95%) and calling the COVID-19 help number (58.3%) were the frequently reported practices among respondents following the development of COVID-19 symptoms. Also, 89 (26%) knew they could contract COVID-19, while 41 (12%) perceived it as an exaggerated event. The effects most frequently reported by respondents were hunger/low income (48.8%) and academic delay (8.8%). Use of face masks by 64.5% and social distancing (48%) were the most frequently reported practices for prevention. Only 71 (20.8%) demonstrated good handwashing practices. The perception of the likelihood to contract COVID-19 and practices to prevent COVID-19 had a weak correlation of 0.239 ($p$ < 0.001).

**Conclusion:** Gaps exist in the practices that prevent COVID-19. There is a need to improve handwashing, use of face masks and other practices that prevent COVID-19. Implications across public health communication and policies were stated.

Corresponding author
Olayinka Ilesanmi,
ileolasteve@yahoo.co.uk

## INTRODUCTION

The Coronavirus infection (COVID-19) is an emerging infectious illness which broke out during the winter of 2019 (*Al-Hanawi et al., 2020*; *WHO, 2020a*). Due to its presentations, it has been declared a public health emergency of international concern by the World Health Organization (WHO) (*WHO, 2020a*). An alarming response has been introduced across the globe due to its high infectiousness and case fatality rate (*Zhong et al., 2020*). The identification of the risks and the prevention of infectivity regarding COVID-19 have been stated to depend on human perception (*Zhong et al., 2020*). Especially in the submergence of an infectious disease such as COVID-19, different thoughts have shaped individuals' views on the illness.

Currently, the Coronavirus disease has spread to 213 countries with nearly 24 million confirmed cases and close to 820,000 recorded deaths (*WHO, 2020b*, *2020c*). Publicly available reports from the Africa Centre for Disease Control (ACDC) states that confirmed cases of COVID-19 had risen to 1,203,769 and 28,289 deaths as of 25 August 2020 (*ACDC, 2020*). As of 25th August 2020, the West African subregion accounted for a significant proportion of cumulative COVID-19 records in Africa. In Nigeria, there are 52,800 confirmed cases of COVID-19 with a total of 1,007 deaths as of 25 August 2020 (*NCDC, 2020a*; *WHO, 2020b*). Oyo State presently holds the third spot on the Nigeria Centre for Disease Control (NCDC) daily COVID-19 updates, with 3058 laboratory-confirmed cases of COVID-19 and 37 deaths (*NCDC, 2020b*). Urban areas in Ibadan, the capital city of Oyo State frequently present with confirmed cases (*Enwongo, 2020*).

As a part of the emergency response activities across all States in Nigeria, health education campaigns have been directed at members of the public (*NCDC, 2020a*, *2020b*). These campaigns have been aimed at knowledge improvement and the correction of certain misconceptions that have been widely circulated among community members (*NCDC, 2020a*). Education on precautionary measures such as wearing of face masks, regular handwashing with soap and water or with alcohol-based hand sanitizers, and social distancing have been done (*NCDC, 2020a*, *2020b*; *Gbadamosi, 2020*).

It is evident that perception shapes one's knowledge and the adoption of safety measures concerning the transmission of an infection. Data obtained from the perception of community members regarding COVID-19 could help target interventions needed to improve the knowledge of community members regarding Coronavirus. Superstitious beliefs have largely shaped the perception of most Nigerians regarding the source and cause of COVID-19 (*Chukwuorji & Iorfa, 2020*). At the onset of the COVID-19 outbreak in Nigeria, infected persons belonged to either the political class or high socioeconomic cadre (*Chukwuorji & Iorfa, 2020*). The characteristic prevalence of COVID-19 infection among this group of persons accorded COVID-19 the name, 'a disease of the rich and mighty' (*Nwaubani, 2020*). Few months into the COVID-19 outbreak in Nigeria, perceptions revolved around "immunity" to COVID-19 among the religious folks with a disregard of bans on religious gatherings (*Lichtenstein, Ajayi & Egbunike, 2020*). Such perceptions could have been influenced by several factors. Social media platforms such as WhatsApp, Facebook and Twitter have been used to spread false news on COVID-19,

resulting to panic disorder and anxiety among some persons and shunning of safety measures among others (*Aluh & Onu, 2020*; *Olapegba et al., 2020*). Among many persons, physical distancing, social isolations, restriction of religious and social gatherings etc. have been opined as alien solutions in overcoming the COVID-19 pandemic in Nigeria and Africa at large (*Olapegba et al., 2020*).

Literatures have reported the existence of knowledge relating to COVID-19 among Nigerians, and it is expected that this would influence precautionary behavior among them. However, inherent wrong perceptions may contribute to COVID-19 risk aversion measures (*Iorfa et al., 2020*). Perceptions of COVID-19 has been influenced by age and gender. Due to their increased vulnerability to illnesses, older persons have been predicted to increasingly adopt COVID-19 precautionary behavior compared to other population groups (*Iorfa et al., 2020*). Females have been identified as models in the adoption of precautionary health behavior. In the COVID-19 context, the practice of handwashing, hygiene, and use of face masks occur more frequently among females than males (*Iorfa et al., 2020*). Such an occurrence could be due to the perceived susceptibility to illnesses among females as well as their health-conscious nature.

Given the importance of risk perception in behavior modification for disease control, it becomes pertinent to assess the perception and practices regarding COVID-19. To the best of our knowledge, the perception and practices of community members in urban areas in Ibadan regarding COVID-19 is currently unknown. An assessment of the perception and practices of community members is important to reduce the risk for COVID-19 infection in Ibadan, a densely populated city in Nigeria. We hypothesized that there is no difference in the sociodemographic characteristics of the community members with the practices of COVID-19 mitigating factors. This study thus aimed at assessing the perception and practices of community members in urban areas in Ibadan regarding COVID-19.

## MATERIALS AND METHODS

### Study design and study setting

A descriptive cross-sectional study design was used. Data was collected using an interviewer-administered questionnaire. Data collection took place from the 3rd to 6th June, 2020. The study was carried out in Ibadan, Oyo State Nigeria. Ibadan is the capital city of Oyo State. Oyo State is one of the states in the south western part of Nigeria. Between 15 June and 10 August, 2020, confirmed COVID-19 cases had risen from 764 to 2,887 in Oyo State, and the State ranks next to Lagos State and the Federal Capital Territory on the NCDC reports for COVID-19 (*NCDC, 2020b*; *Enwongo, 2020*). The official language in Nigeria is English, while the major informal language for communication in Ibadan is Yoruba, which has different dialects.

### Study population

The study population for the survey was one eligible member of the households in the selected urban communities in Ibadan, Oyo State. All consenting household members were

included in the study. Household members that were less than 18 years were excluded. Verbal consent was obtained from participants.

## Sample size determination and sampling technique

The sample size was calculated using sample size formula for descriptive cross-sectional study. The population of the selected LGA is >100,000. The sample size was calculated using the Leslie Kish formula for sample size determination for a single proportion as follows:

$n = Z\alpha^2 p(1 - p)/d^2$ where:

$n$ = Minimum desired sample size

$Z$ = the standard normal deviate, usually set as 1.96 which corresponds to 5% level of significance.

$P$ = 50% was be used

$d$ = Degree of accuracy (precision) set at 5% (0.05)

$n = 1.96^2 \times 0.5 \times (1 - 0.5)/0.05^2 = 384$

A sample of 360 (93.8%) were studied in the urban communities of Ibadan.

A multi-stage sampling technique was used to select the respondents for the study

Stage 1:

Simple random sampling was used to select 3 out of the 6 urban local government area in Ibadan.

Stage 2:

In each of the selected LGA, a political ward was chosen for the study.

Stage 3:

A center location was chosen in the selected ward. A bottle was rotated to determine the direction of movement of the interviewers. From the direction of the bottle tip all consenting eligible adults from the households were included in the study until 120 persons were interviewed in each LGA.

Sampling of 120 each in the three urban LGA gives a total sample size of 360.

## Data collection methods

The questionnaire has two sections.

Section A: sociodemographic characteristics

The sociodemographic characteristics include age of respondents, sex, highest level of education, ethnicity and occupation.

Section B: perception and practices regarding COVID-19.

Close-ended questions were asked on perception of the respondents on COVID-19, their current practices, and what they would do if they were infected. Open-ended questions were asked on the effects of COVID-19 on and suggestions to the government to curb the pandemic.

A six-point question was asked on the practice of handwashing. The respondents were asked to demonstrate how they usually practice handwashing. The interviewer correctly marked all the points demonstrated by respondents.
The questionnaire was adapted from a tool used for a similar perception study on Ebola Virus Disease in 2014 (*Gidado et al., 2015*). The tool was validated by an infectious disease epidemiologist. Pre-testing of the tool was done by administering 10 questionnaires in another Local Government Area not selected for the study. A few ambiguous questions were modified. Back-to-back translation of the questionnaire was done by experts who had sound understanding of the Yoruba language. The questionnaire was administered to most of the respondents in Yoruba Language. Data collection was done by trained research assistants with a minimum of first degree.

Independent variables included: Sociodemographic characteristics like age, sex, level of education, and occupation.

Outcome/dependent variables were the practice of handwashing and the use of other mitigating measures.

## Data management

Data were analyzed with SPSS version 23. Age was summarized using mean and standard deviation, while frequencies and percentages were used for categorical variables. A total score of 6 was assigned to good practice of handwashing after the respondents were asked to demonstrate hand washing. One point each was assigned for the following: palm, back of the hand, spaces between the fingers, fingernails, wrist and thumbs. Only those who demonstrated the six points were categorized to have had a good practice of handwashing. Chi-square test was used for the assessment of associations between sociodemographic characteristics and practice of handwashing. Pearson correlation was conducted between the perception of the likelihood of contracting COVID-19 and practices to prevent COVID-19. Multivariate analysis of the determinants of good handwashing practices was conducted using Logistic regression. $P$-values < 0.05 were accepted as significant.

## Ethical approval and consent to participate

Ethical approval to carry out the study was obtained from the Oyo State Ministry of Health Ethical Review Committee, with reference number AD/13/479/1779[A]. Permission for the study was sought from the respondents and the confidentiality of information was ensured. The respondents were informed of their right to decline or withdraw from the study at any time without any adverse consequences. No harm was inflicted on participants because of participation in this study.

## RESULTS

A total of 360 respondents were interviewed among urban residents in Ibadan. The mean age was 33.2 ± 10.6 years and 225 (62.5%) were females. Those with secondary education and above were 332 (92.2%), 314 (87.2%) were of the Yoruba ethnic group, and 171 (47.5%) engaged in business or trading (Table 1). Among the 360 respondents 342 (95%) have heard of COVID-19.

Most frequently reported practices among respondents following the development of COVID-19 symptoms were: Going to the hospital 171 (50%) and calling the COVID-19

**Table 1  Sociodemographic characteristics of respondents among Ibadan residents, 2020.**

| Socio-demographic characteristics | Frequency | % |
|---|---|---|
| Age group (Years) | | |
| <25 | 70 | 19.4 |
| 25–34 | 136 | 37.8 |
| 35–44 | 106 | 29.4 |
| ≥45 | 48 | 13.3 |
| Sex | | |
| Male | 135 | 37.5 |
| Female | 225 | 62.5 |
| Highest level of education | | |
| Primary and below | 28 | 7.8 |
| Secondary and above | 332 | 92.2 |
| Ethnicity | | |
| Yoruba | 314 | 87.2 |
| Ibo | 31 | 8.6 |
| Hausa | 8 | 2.2 |
| Others | 7 | 1.9 |
| Occupation | | |
| Business/trader | 171 | 47.5 |
| Artisans | 110 | 30.6 |
| Professional/civil servant | 30 | 8.3 |
| Unemployed/housewife/student | 49 | 13.6 |

help number 105 (30.7%). The other reported practices included: praying and staying at home each with 29 (8.5%) respondents as shown in Fig. 1.

Regarding COVID-19 risk perception, 89 (26%) knew they could contract COVID-19, while 41 (12%) perceived it as an exaggerated event. It was also perceived as an intention for corruption by 23 (6.7%), COVID-19 was an attack by the Western World was reported by 68 (19.9%) and 122 (35.7%) called COVID-19 a source of panic. The effects most frequently reported by respondents were hunger/low income among 167 (48.8%) and academic delay among 30 (8.8%). Regarding suggestions to the government, 108 (31.6%) suggested the provision of medical supplies/palliatives/seeking of cure, while 68 (19.9%) suggested free testing/free treatment. Other effects of COVID-19 and suggestions to the government are as shown in Table 2.

The most frequently reported practice for prevention of COVID-19 among respondents were the use of face masks by 224 (65.5%) and social distancing by 164 (48%). Others included: staying at home/following COVID-19 updates 8 (2.2%), taking Vitamin C/fruits/warm water 4 (1.1%), and doing nothing 5 (1.4%) as shown in Fig. 2.

Figure 3 shows that only 80 (22%) of respondents demonstrated good handwashing practices. Among respondents aged less than 25 years, 16 (23.5%) had good handwashing practices compared to 14 (29.8%) aged above 45 years. Among females, 49 (22.8%) had

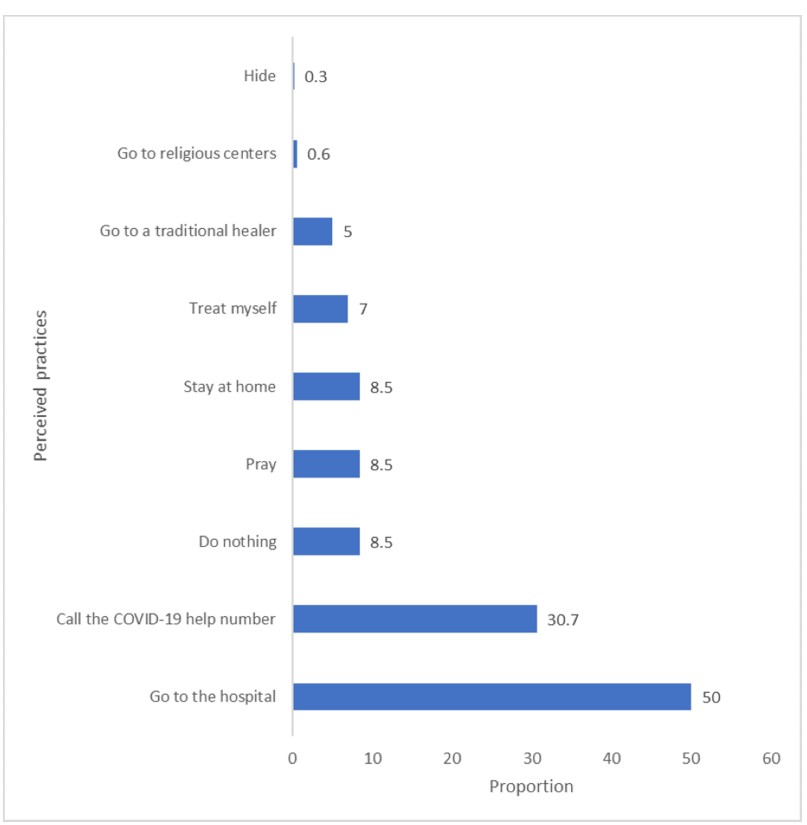

**Figure 1 Practices of Ibadan residents following the development of COVID-19 symptoms.**

good handwashing practices compared to 22 (17.3%) males although these differences are not statistically significant (Table 3).

Males have 27.5% less odds of having good hand washing practice compared to females, though not statistically significant (AOR 0.725, 95% CI [0.418–1.259], $p = 0.253$) (Table 4).

The perception of the likelihood to contract COVID-19 and practices to prevent COVID-19 had a weak positive correlation of 0.239 ($p < 0.001$).

## DISCUSSION

This study found that many individuals lived in denial of the existence of COVID-19. The perception of the illness as an avenue for politicians to enrich themselves indicates that there still exists inadequate knowledge of COVID-19 among community members in Ibadan. Denial, ignorance regarding COVID-19, and the existing lack of trust in the Nigerian government have been reported since the outbreak of COVID-19 in Nigeria (*Chukwuorji & Iorfa, 2020*). From the present study, a high rating of the perceived likelihood of contracting COVID-19 was observed among 26% of respondents, while it was minimally perceived as an attack by the Western World among nearly 20%.

Findings obtained from this study revealed that the practices most often adopted following the development of COVID-19 symptoms were either to go to the hospital or call the COVID-19 help number. This indicates that the source of help for COVID-19

**Table 2 Perceptions and effects of COVID-19 and suggestions to government by community members in Ibadan, 2020.**

| Variables | n (%) |
| --- | --- |
| Perception on COVID-19 | |
| It creates a lot of panic | 122 (35.7) |
| It is a deadly disease | 94 (27.5) |
| I am at risk of COVID-19 infection | 89 (26) |
| It is highly infectious | 72 (21.1) |
| It is an attack by the Western World | 68 (19.9) |
| It is just being exaggerated | 41 (12) |
| It has no cure | 33 (9.6) |
| Don't believe it exists | 28 (8.2) |
| An intention for corruption | 23 (6.7) |
| Effects of COVID-19 | |
| Hunger/low income | 167 (48.8) |
| Academic delay | 30 (8.8) |
| Restricted movement/no going to work | 25 (7.3) |
| No gatherings | 20 (5.8) |
| Suggestions to government | |
| Provide medical supplies/palliatives/seek cure | 108 (31.6) |
| Health education/enforce preventive measures | 70 (20.5) |
| Free testing/free treatment | 68 (19.9) |
| Stop reporting false figures/lift lockdown and bans | 44 (12.9) |
| No idea/do anything | 27 (7.9) |

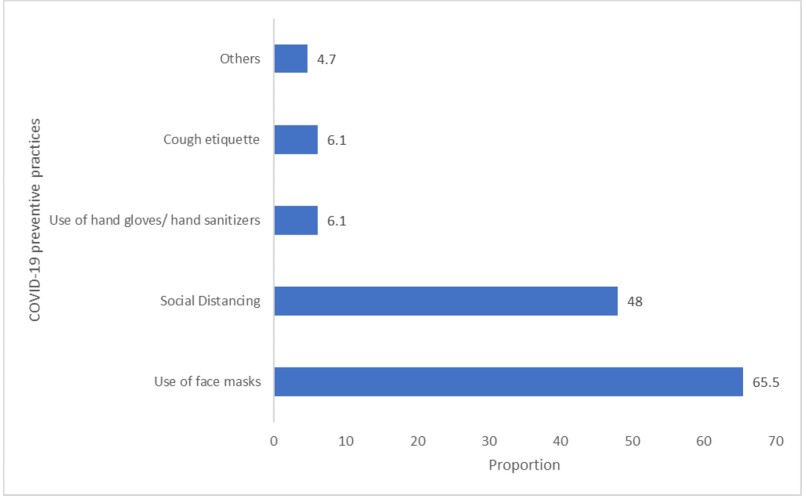

**Figure 2 Practices of COVID-19 prevention among respondents.**

treatment is well known among community members in urban areas of Ibadan. Although distrust in government capacity regarding COVID-19 is currently obtained, individuals are willing to take proactive measures following the suspected development of COVID-19

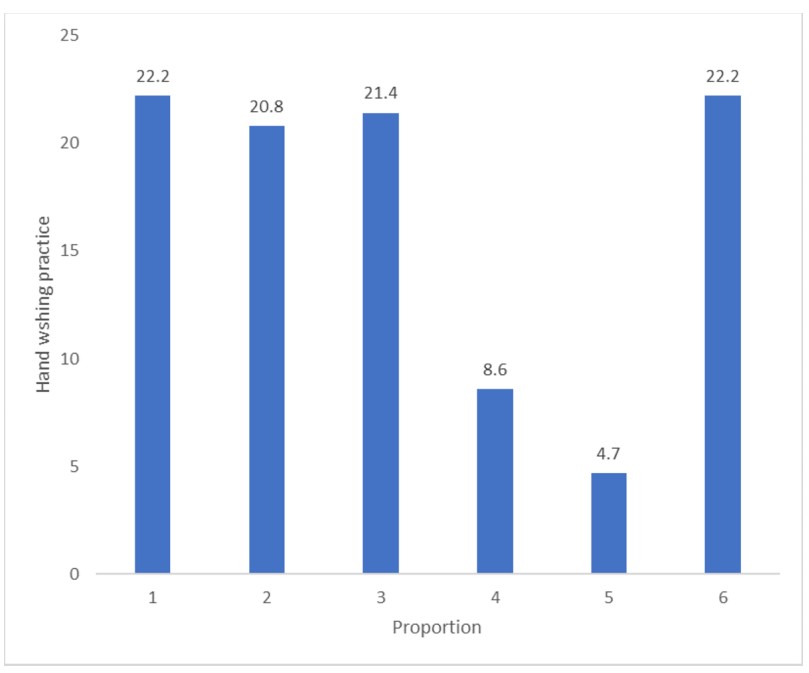

**Figure 3** **Points scored in handwashing demonstration.**

symptoms (*Chukwuorji & Iorfa, 2020*). An Indian study similarly reported that hospital visitation was frequently opted for as a step to be taken following the development of COVID-19 in individuals in a close relationship (*Dkhar et al., 2020*).

We found that the use of face masks and practice of social distancing measures were more frequently embraced among respondents compared to other COVID-19 mitigation measures, although full adherence was low. A web-based study conducted in Nigeria mostly stated mouth-covering while sneezing, wearing of face masks and avoidance of crowded spaces as self-reported practices among respondents (*Iorfa et al., 2020*). Our findings revealed that myriads of perceptions were associated with COVID-19. These included COVID-19 as an exaggerated illness with intentions for corruption, its highly infectious and deadly nature, and a reason for panic disorders. Similarly, the likelihood of positive practices concerning COVID-19 was associated with a positive perception of the risk of infection (*Zhong et al., 2020*). Findings from previous studies conducted in Nigeria also corroborate the key role of positive risk perception on imbibing COVID-19 protective practices and attitudes (*Iorfa et al., 2020*). The finding from the present study however contradicts the assumption of the Health Belief Model (HBM) that protective actions are more likely to succeed a high level of perceived susceptibility (*Tarkang & Zotor, 2015*). The results obtained herein is higher than the knowledge concerning the practice of face masks in Saudi Arabia (*Al-Hanawi et al., 2020*). Due to its deadly nature, COVID-19 has introduced fear which has compelled protective actions from individuals regarding the illness (*Zhong et al., 2020*).

Previous studies have shown that fear could motivate healthy behavior among individuals especially during epidemics, but such behavior may not be sustainable (*Witte, 1998*; *Nabi, 1999*; *Ufuwa et al., 2020*). The adoption of these healthy behaviors in
**Table 3 Association between sociodemographic variables and practice of handwashing among community members who have heard of COVID-19 in Ibadan 2020.**

| Sociodemographic variable | Practice of handwashing | | Chi-square | *p*-Value |
|---|---|---|---|---|
| | Good *n* (%) | Poor *n* (%) | | |
| Age | | | | |
| <25 | 16 (23.5) | 52 (76.5) | 3.890 | 0.274 |
| 25–34 | 22 (16.9) | 108 (83.1) | | |
| 35–44 | 19 (19.6) | 78 (80.4) | | |
| >44 | 14 (29.8) | 33 (70.2) | | |
| Sex | | | | |
| Male | 22 (17.3) | 105 (82.7) | 1.451 | 0.228 |
| Female | 49 (22.8) | 166 (77.2) | | |
| Highest level of education | | | | |
| Primary and below | 7 (26.9) | 19 (73.1) | 1.109 | 0.775 |
| Secondary and above | 64 (20.3) | 252 (79.7) | | |
| Ethnicity | | | | |
| Yoruba | 62 (20.8) | 236 (79.2) | 0.592 | 0.898 |
| Ibo | 6 (20.7) | 23 (79.3) | | |
| Hausa | 1 (12.5) | 7 (87.5) | | |
| Others | 2 (28.6) | 5 (71.4) | | |
| Occupation | | | | |
| Business/trader | 31 (19.3) | 130 (80.7) | 0.915 | 0.822 |
| Artisans | 24 (23.1) | 80 (76.9) | | |
| Professional/civil servant | 5 (17.2) | 24 (82.8) | | |
| Unemployed/housewife/student | 11 (32.9) | 37 (77.1) | | |

the present study is in tandem with the recommendations of the World Health Organization (WHO) on safety measures for COVID-19 (*WHO, 2020c*). The insufficiency of fear as a propellant for adherence to recommended guidelines for COVID-19 has been reported to be an outplay of knowledge-attitude discrepancy (*Iorfa et al., 2020*). These findings imply that individual perception of infectious illnesses such as COVID-19 may not be sufficient to influence the adoption of protective practices. This explains the need for the regular sensitization of community members on COVID-19 safety measures regardless of their perception concerning the illness.

We found that the practice of handwashing was commoner among individuals with a greater risk perception for COVID-19. Because these individuals perceive themselves as vulnerable to COVID-19 infection, they are more likely to engage in handwashing practice. Handwashing practice has been identified as one of the mitigation strategies for breaking the chain of COVID-19 transmission. An online-based Nigerian survey revealed a higher practice of handwashing compared to other COVID-19 preventive measures (*Iorfa et al., 2020*). A study conducted in Ibadan on hand hygiene practices post the Ebola virus disease outbreak revealed a high proportion of inadequate self-reported hand hygiene

**Table 4 Multivariate analysis of the determinants of good handwashing practices.**

| Sociodemographic variable | AOR | 95% CI of AOR | | p-Value |
|---|---|---|---|---|
| | | Lower | Upper | |
| Age | | | | |
| <25 | 0.764 | 0.276 | 2.116 | 0.605 |
| 25–34 | 0.534 | 0.248 | 1.151 | 0.109 |
| 35–44 | 0.595 | 0.271 | 1.306 | 0.196 |
| >44 | 1 | | | |
| Sex | | | | |
| Male | 0.725 | 0.418 | 1.259 | 0.253 |
| Female | 1 | | | |
| Highest level of education | | | | |
| Primary and below | 1.146 | 0.451 | 2.911 | 0.775 |
| Secondary and above | | | | |
| Ethnicity | | | | |
| Yoruba | 1.279 | 0.534 | 3.065 | 0.581 |
| Ibo | 0.750 | 0.083 | 6.735 | 0.797 |
| Hausa | 1.279 | 0.534 | 3.065 | 0.581 |
| Others | | | | |
| Occupation | | | | |
| Business/trader | 0.933 | 0.358 | 2.434 | 0.888 |
| Artisans | 1.619 | 0.546 | 4.804 | 0.385 |
| Professional/civil servant | 0.869 | 0.219 | 3.448 | 0.842 |
| Unemployed/housewife/student | 1 | | | |

practice (*Martins & Osiyemi, 2017*). Lassa fever studies conducted in Edo State reported inadequate handwashing practices, while a similar study in Kaduna State, Nigeria reported good handwashing practices among respondents (*Tobin et al., 2013*). The similarities of most of these findings with ours imply the wide acceptance of the practice of handwashing in the management of infectious diseases.

Findings from this study revealed a higher likelihood of good handwashing practices among females than males, although it was not significant. Our finding contradicts cultural notions which suggests that hygiene measures are more frequently practiced among females than males. However, a few other studies have reported no difference in the practice of hand hygiene among males and females in Nigeria (*Ogunsola et al., 2013*; *Martins & Osiyemi, 2017*). The agreement of our findings with reference literatures could be due to the alienation of regular and proper handwashing practices in the Nigerian context. This could therefore have contributed to the observed level of handwashing practices among males and females as found in this study. The availability of water and sanitation access have been identified as major determinants of good handwashing practices (*Ogunsola et al., 2013*). However, these basic amenities are not readily available in many Nigerian homes (*Uchejeso & Obiora, 2020*). This therefore prompts overcrowding of persons at wells and boreholes, a condition which necessitates the use of water in

small amounts either for handwashing or other purposes. Findings from this study thus imply the need for improved access and portable water supply as required for the reduction of COVID-19 transmission in the communities.

We found that COVID-19 poses significant threat to local economy, resulting in low income and resultant hunger. This is likely due to the increased cost of purchasing goods or a result of the lockdown which has denied many individuals the opportunity to earn their income. Denial of opportunities to engage in money-making ventures was experienced and impacts such as hunger was greatly felt among many persons (*Chukwuorji & Iorfa, 2020*). This explains the need for the provision of palliatives to fight hunger and reduce susceptibility to other infections during the COVID-19 outbreak. Similarly, decreased productivity and job losses and an unprecedented economic disaster have been reported (*Atalan, 2020*). Contrary to the finding in this study, other studies have reported stress and anxiety as psychological reactions due to the Coronavirus pandemic (*Atalan, 2020*). Other psychological reactions such as boredom, anger, and loneliness have been notably identified as resultant threats during the COVID-19 pandemic (*Aluh & Onu, 2020*). This calls for the provision of psychosocial support for individuals during the COVID-19 lockdown. Interestingly, a recognition of the significance of essential staff has also resulted from the COVID-19 outbreak (*Spowart, 2020*).

Pertaining to suggestions to the government concerning COVID-19 containment, the provision of medical supplies and palliatives received highest recommendation among respondents. Most Nigerian households depend on daily earnings of breadwinners, and difficulty in survival was experienced during the COVID-19 lockdown which lasted for three months in Nigeria (*Chukwuorji & Iorfa, 2020*). Also, health education, the enforcement of preventive measures, and free testing and treatment received much recognition. These imply two things. Firstly, health education campaigns concerning COVID-19 should be conducted by public health officials in simple, unambiguous languages which will facilitate the understanding of community members. Secondly, the availability of medical supplies and palliatives would enhance the adherence to safety measures for COVID-19, such as the use of face masks among community members. Similar suggestions have been made in previous studies (*Kebede et al., 2020*).

## Strengths of the study

Up to date, most studies on perception and practices regarding COVID-19 have used electronic sources for data collection, and such results may have been biased. Our study is a community-based physical study that used a semi-structured interviewer-administered questionnaire. To the best of our knowledge, it is the first to study the perception and practices of adult population in urban communities in Nigeria. The study also made use of an adequate sample size (360 adults).

## Limitations of the study

As this study was limited to the perception and practices regarding COVID-19, the knowledge of community members on the illness was not addressed. The assessment of

factors influencing COVID-19 practices among community members was obscure in this study.

## CONCLUSIONS

The adoption of preventive measures is critical to forestall onward transmission of COVID-19. However, adequate, and correct risk perception for COVID-19 is required to enable the adoption of COVID-19 safety measures. We hereby recommend enhanced sensitization and health education sessions for all community members about COVID-19 in Ibadan metropolis regardless of their sociodemographic characteristics. Also, health campaigns should be more focused on practices such as regular handwashing with soap and water and social distancing, which protect against transmission of COVID-19 among community members irrespective of their sex. In addition, access of individuals to portable source of water supply should be enabled by increased provision of water sources in residential apartments. The government should also install more infrastructures for water supply where dearth of water exists.

## ACKNOWLEDGEMENTS

The authors express their gratitude to all community members for their willingness and cooperation to participate in this study.

### Funding
The authors received no funding for this work.

### Competing Interests
The authors declare no competing interests.

### Author Contributions
- Olayinka Ilesanmi conceived and designed the experiments, performed the experiments, analyzed the data, prepared figures and/or tables, authored or reviewed drafts of the paper, and approved the final draft.
- Aanuoluwapo Afolabi performed the experiments, analyzed the data, prepared figures and/or tables, authored or reviewed drafts of the paper, and approved the final draft.

### Human Ethics
The following information was supplied relating to ethical approvals (i.e., approving body and any reference numbers):

Ethical approval to carry out the study was obtained from the Oyo State Ministry of Health Ethical Review Committee, with reference number AD/13/479/1779A.

### Data Availability
Raw data is available as a Supplemental File.

## Supplemental Information

Supplemental information for this article can be found online at http://dx.doi.org/10.7717/peerj.10038#supplemental-information.

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
