# Peer review of "Perception and practices during the COVID-19 pandemic in an urban community in Nigeria: a cross-sectional study"

_PeerJ, doi:10.7717/peerj.10038_

## Round 0.1 · original submission · Major Revisions

The manuscript (MS) reports on an important study describing people's perceptions regarding COVID19 and how they wash their hands in addition to other things. In addition to the issues raised by the reviewers, please also address the following:

1- Abstract: In Methods: add date, location, inclusion criteria, outcome, exposure? how was data collected? explain what the bivariate analysis included...Add the number of participants and edit the Conclusion so that it is aligned to aim, methods and results.

2- Introduction: L63: what does "NCDC" mean?, L75-77 and L94-95: remove this, it is repeated from before.

3- Methods: L105-107: delete this part, it is repeated. I also agree with the Reviewer about the need for more details about the questionnaire, number and type of questions, questionnaire development and validation and testing- which language was it used in? What software was used for analysis.

4- Results: Analysis plan: L230-232 and elsewhere: the claim that risk perception is associated with the adoption of preventive practices is not supported by the current analysis. The association between the two factors was not assessed in the last table along with the other variables and thus must be addressed. Another important point to address is that the cross sectional design of the study means that the study has lots of confounders that should be controlled with regression analysis instead of the bivariate analysis done in the study. Please address this too. Regression and controlling the confounders will most likely change the observed bivariate associations reported now.

5- Discussion: respond to the aim in the opening statement of discussion. In L208 and also mentioned before, what does "motive for corruption" mean? L210-211, L217-219, L293-294: remove this repeated part. Compare the findings to COVID19 studies, not to SARS or Ebola studies. L278: what does "essential staff" mean?"

6- Conclusion: L311: pls rephrase to clarify this sentence. It is not clear.

7- Figure 1: how are these "perceived" practices? calculate and display percentages not numbers. Do this also in Figures 2 and 3.

8- Table 1, last row: there were suggestions for the government to pray? Please clarify

Reviewer 1 ·

Basic reporting

The study is clear, unambiguous, professional English language used throughout. The literature used well referenced & relevant. The structure conforms to PeerJ standards, the raw data available.

Experimental design

The methodology of the study should adhere to STROBE statement for reporting cross-sectional studies. The authors could make use of this checklist to improve this manuscript. Please follow this link: https://www.strobe-statement.org/fileadmin/Strobe/uploads/checklists/STROBE_checklist_v4_cross-sectional.pdf

- Please mention in details the formula used to calculate the sample size and your population size.
- Explain more about the tool development, and the references you have used during the development.
- You need to add section about the validation of the survey. Have you conducted a pilot study etc, panel of expert review, have you modified some questions based on the feedback received during pilot study? Face validity, content validity.....

Validity of the findings

The authors developed the questionnaire and did not mention any thing about the validity. Content validity usually used for such questionnaire development . Perhaps the authors could elaborate the content validity procedure in a detailed paragraph in methods as well as in results. The authors could mention content validity index and content validity ratio to improve the rigor of method. I have provided link to content validity literature from LAWSHE 1975 below:
http://citeseerx.ist.psu.edu/viewdoc/download?doi=10.1.1.460.9380&rep=rep1&type=pdf
I have also provided a research paper that has used content validity method and interpreted the findings. Please read the study and try to report the content validity in the manner reported by the study. Please see the link: https://www.ncbi.nlm.nih.gov/pmc/articles/PMC5750538/
You may wish to cite this study as a backup to your content validity method selection.

Additional comments

This manuscript highlights an important area of research that is currently booming all over the world. I have had pleasure in reading this manuscript and have recommended important modifications that I believe are essential to improve the manuscript's scientific rigor and worthiness for publication.
The introduction should logically progress to the purpose statement. The analogy of a funnel is helpful when thinking of the introduction, which follows a logical sequence from broadly presenting the concept to focusing on a specific knowledge gap, discrepancy between studies, or research question that leads to the purpose statement. In other words, Mention that explicitly and describe why it is important to learn from this region. The introduction and discussion section could benefit from recent study about COVID-19 public perceptions which was conducted in Arab countries including Egypt which located in Africa. This study would also help in formulating the introduction and discussion of this paper.
https://joppp.biomedcentral.com/articles/10.1186/s40545-020-00247-x

- The methodology of the study should adhere to STROBE statement for reporting cross-sectional studies. The authors could make use of this checklist to improve this manuscript. Please follow this link:
https://www.strobe-statement.org/fileadmin/Strobe/uploads/checklists/STROBE_checklist_v4_cross-sectional.pdf

- Please mention in details the formula used to calculate the sample size and your population size.
- Explain more about the tool development, and the references you have used during the development.
- You need to add section about the validation of the survey. Have you conducted a pilot study etc, panel of expert review, have you modified some questions based on the feedback received during pilot study? Face validity, content validity.....
The authors developed the questionnaire and did not mention any thing about the validity. Content validity usually used for such questionnaire development . Perhaps the authors could elaborate the content validity procedure in a detailed paragraph in methods as well as in results. The authors could mention content validity index and content validity ratio to improve the rigor of method. I have provided link to content validity literature from LAWSHE 1975 below:
http://citeseerx.ist.psu.edu/viewdoc/download?doi=10.1.1.460.9380&rep=rep1&type=pdf
I have also provided a research paper that has used content validity method and interpreted the findings. Please read the study and try to report the content validity in the manner reported by the study. Please see the link:
https://www.ncbi.nlm.nih.gov/pmc/articles/PMC5750538/

You may wish to cite this study as a backup to your content validity method selection.

"Moslem and Muslim are basically two different spellings for the same word." But the seemingly arbitrary choice of spellings is a sensitive subject for many followers of Islam. Whereas for most English speakers, the two words are synonymous in meaning, the Arabic roots of the two words are very different. A Muslim in Arabic means “one who gives himself to God," and is by definition, someone who adheres to Islam. By contrast, a Moslem in Arabic means “one who is evil and unjust" when the word is pronounced, as it is in English, Mozlem with a z. Muslim is more preferred by scholars and by English-speaking adherents of Islam.
Regarding your finding about Muslim, All the 6 points used for scoring is done by Muslims 5 times a day during wudu steps which performed before prayers. Please watch this short clip.
https://www.youtube.com/watch?v=F1AyRBejDVk&t=125s
Therefore I think this finding may affect the validity of your result as it is against the logic. The finding could be during to reporting bias by people who collected the data. My advise to you is to delete the religion socio-demographic variable part from your result and discussion and abstract.

·

Basic reporting

English used is acceptable, quite clear and unambiguous. However, a general editing for punctuation errors should be conducted.

Background context and literature are not sufficient. Please check the general comments for further suggestions to improve this aspect of your manuscript.

Hypotheses were not clearly stated. You may want to look into this.

Experimental design

Research questions are well defined and research gap filled is well explained.

The investigations performed are not rigorous enough. Further statistical analysis may be conducted to improve the quality of the manuscript. Check the general comments section to see further suggestions on how to go about this.

Method is fairly described. The authors can do more.

Validity of the findings

Fair.

Please check the general comments section on suggestions to better discuss your findings.

Additional comments

The authors have conducted an important study which is relevant and timely. The manuscript will however benefit from the following observations

Title
Include a definite article “the” before “Covid-19 in the title, such that it reads, “... during the COVID-19 pandemic...”

Abstract
In the Abstract, under the method section, put a full stop (.) after the first sentence.
State the number of household members sampled in the method section of the Abstract as well as their gender distribution and their mean age or age range as you may desire. Remove the statement on age from the results section. You didn’t set out to find out their ages as your primary hypothesis, so state it in your description of demographics under the method section.

Introduction
Page 1, Paragraph 1, Line 12 of the Introduction: Please change “individual’s” to “ individuals’ “
Please update COVID-19 case numbers to reflect the current status as at the date the manuscript will be resubmitted.
Check for the proper use of punctuations all through the work. The punctuation mark, full stop (.) is neglected in several places through out the manuscript.
The literature review conducted by the authors is not rich enough. A Covid-19 related study in their home country should leverage on some already published empirical findings surrounding their interest from the same locality. I’d suggest they review the following works freely available online and cite appropriately. This will improve the quality of their literature review;
Chukwuorji, J. C., & Iorfa, S. K. (2020). Commentary on the coronavirus pandemic: Nigeria. Psychological Trauma: Theory, Research, Practice, and Policy, 12(S1), S188-S190. http://dx.doi.org/10.1037/tra0000786

Iorfa, S.K., Ottu, I.F.A., et al (2020) COVID-19 knowledge, risk perception and precautionary behaviour among Nigerians: A moderated mediation approach. https://doi.org/10.1101/2020.05.20.20104786

Olapegba, P., Ayandele, O., et al (2020)A Preliminary Assessment of Novel Coronavirus (COVID-19) Knowledge and Perceptions in Nigeria. https://doi.org/10.1101/2020.04.11.20061408

Aluh, D. O., & Onu, J. U. (2020). The need for psychosocial support amid COVID-19 crises in Nigeria. Psychological Trauma: Theory, Research, Practice, and Policy, 12(5), 557–558. https://doi.org/10.1037/tra0000704

Methods
Please insert “of” between the dates, e.g, 3rd of June...
Update case numbers and rank for Oyo State before resubmission.
The whole of the methodology section needs to be rewritten in line with the journal guidelines. Remove unnecessary information and include very important details about your sample/participants characteristics. Ensure that it is okay to report demographics under results instead of adding it up as sample characteristics.
Results
Your analysis is too basic. Could you not at least conduct some correlation analysis to see the relationships between some of your variables? It would be interesting to find out how those perceptions correlate with practices.

Discussions
Compare and congrats your findings with others conducted in Nigeria as well and not just those in Ethiopia and other countries. Take a look at Olapegba et al (2020) and Iorfa et al (2020) stated earlier.

Conclusions
Your conclusion is vague and ambiguous. Could you be more specific in giving practical recommendations? For instance, when you speak of practices which protect against transmission...., what practices are you referring to? Could you please give examples?

References
Please format all your references accordingly to fit the journal format.

---

## Round 0.2 · Major Revisions

The manuscript provides a useful description of an interesting study. The reason I marked my decision as "Major" instead of "Minor" is to indicate that the changes must be done:

1- This is a cross-sectional study with a good sampling strategy and several collected variables. Cross-sectional studies are prone to confounders. Conduct a regression analysis to support the bivariate analysis that already exists.

2- Read the paper carefully, remove the existing track changes, and make sure that parts (such as those related to the questionnaire origin and validity) are not repeated in several places in the paper. Please make this throughout the paper from Introduction to Discussion. Edit the new parts you added in this version and fit them into the narrative aiming for brevity and relevance. Relevance indicates that you remove parts not related to the study point such as those about Ebola and other diseases, especially after the addition of the new references about COVID-19.

Reviewer 1 ·

Basic reporting

This version is much better than precious one. The author had addressed the majority of requested comments.

Experimental design

Valid Experimental design

Validity of the findings

valid

Additional comments

Thank you for addressing the majority of the comments

·

Basic reporting

No comment

Experimental design

No comment

Validity of the findings

No comment

Additional comments

I thank the author(s) for their diligent responses to the reviewer comments. The manuscript has greatly improved. I have only a few minor corrections to point out. Do not be scared by the length of the additions, they are only minor corrections. Congratulations.
In the abstract and elsewhere in the manuscript, change your usage of "analysis was done" to “analysis was conducted" or "analyses were conducted" as the case may be.
In the abstract, first state the total number of respondents in the methods section. There will therefore be no need to repeat this information in the results section
Try a statement like this;
A descriptive cross-sectional study design using a multistage sampling technique was used to recruit 360 respondents (Mean age; 33.2; 62.5% females) from households in....
If you have done this, there will be no need to capture the educational qualification here in the methods section or in the result section.
Please state how long the data collection took. For instance, instead of “data was collected using an interviewer-administered questionnaire in June 2020.” you may replace with “data was collected using an interviewer-administered questionnaire from 3rd-6th June, 2020..
Remove the sentence that begins with “Household members…”
In sum, if you accept my suggestions, the method section of your abstract should read something like this
A descriptive cross-sectional study design using a multistage sampling technique was used to recruit 360 respondents (Mean age; 33.2; 62.5% females) from households in Ibadan. Data was collected using an interviewer-administered questionnaire from 3rd-6th June, 2020. Those who demonstrated washing of the palm, back of the hand, spaces between the fingers, fingernails, wrist, and thumbs had 6 points and were categorized to have had a good practice of hand washing. Descriptive statistics were conducted. Bivariate analyses of sociodemographic characteristics and good hand washing practices were conducted using Chi square tests. P values <0.05 were statistically significant.
Please note the additions of “s” and the changes from “analysis” to “analyses”
The results section should read like this;
Going to the hospital (95%) and calling the COVID-19 help number (58.3%) were the frequently reported practices among respondents following the development of COVID-19 symptoms. Also, 89 (26%) knew they could contract COVID-19, while 41 (12%) perceived it as an exaggerated event. The effects most frequently reported by respondents were hunger/low income (48.8%) and academic delay (8.8%). Use of face masks by 64.5% and social distancing (48%) were the most frequently reported practices for prevention. Only 71 (20.8%) demonstrated good hand washing practices. The perception of likelihood to contract COVID-19 and practices to prevent COVID-19 had a weak correlation of 0.239(p<0.001).
Please note the change from “signs” to “symptoms”. Effect this everywhere it appears in the body of the manuscript. Notice the inclusion of parenthesis for the 48%.
Include in your conclusion, a brief statement on the implications of the findings. If you love this example, you may use it. “Implications across public health communication and policies were stated.” If you have included this, then include a little policy implications and implications for public health communication and awareness campaigns in your discussions. Okay, I see you have alluded to this in the discussions already. You may add to it if you want to.
Remember to update case numbers for COVID-19 before re-submitting the manuscript.
Page 3, Line 75: Note the repetition of “in”. Please remove one.
Page 3, Line 76: Change the citation for “The Whistler, 2020” to “Enwongo, 2020”. Do this every other place it appears in the manuscript inorder not to have conflicting information in the citation and references.
Page 4, Line 114: Remove “The use of” and let the sentence read as “Social media platforms such as…”
You have provided a section for study instruments separately and another section for Data collection methods and instruments. Please merge these appropriately. Provide reliability information for the Yoruba and English versions of your adapted scale. Cronbach alphas are enough.
Page 10, Line 289: Change “poor” to “inadequate”

---

## Round 0.3 · Minor Revisions

The addition of the multivariable analysis provides a correct perspective about the association between handwashing and personal background which the author chose to associate with the dependent variable. Statistical significance is not the critical factors to determine whether to keep or drop the multivariable regression. Rather, this depends on the study design.

I previously asked the authors to thoroughly check the paper and make sure repeated parts are removed. I am attaching a marked pdf file with required modifications as follows:
1- Three areas were repeated parts/ typos need to be removed.
2- Modification to add the multivariable analysis to the descripton of methods, another one to emphasize its results in Discussion and a third one to state its implication in Conclusion.
3- Two modifications to language-edit some parts.

Please implement these changes.

---

## Round 0.4 · accepted · Accept

Most of the required modifications were applied.